# Comparative Analysis of Vision Transformer Models for Facial Emotion Recognition Using Augmented Balanced Datasets

**Sukhrob Bobojanov [1], Byeong Man Kim [1], Mukhriddin Arabboev [2,*] and Shohruh Begmatov [2]**

1   Computer Software Engineering, Kumoh National Institute of Technology, Gumi 39177, Republic of Korea;
    w.suxrob.w@gmail.com (S.B.); bmkim@kumoh.ac.kr (B.M.K.)
2   Tashkent University of Information Technologies named after Muhammad al-Khwarizmi, Tashkent 10084,
    Uzbekistan; bek.shohruh@gmail.com
*   Correspondence: mukhriddin.9207@gmail.com

**Abstract:** Facial emotion recognition (FER) has a huge importance in the field of human–machine interface. Given the intricacies of human facial expressions and the inherent variations in images, which are characterized by diverse facial poses and lighting conditions, the task of FER remains a challenging endeavour for computer-based models. Recent advancements have seen vision transformer (ViT) models attain state-of-the-art results across various computer vision tasks, encompassing image classification, object detection, and segmentation. Moreover, one of the most important aspects of creating strong machine learning models is correcting data imbalances. To avoid biased predictions and guarantee reliable findings, it is essential to maintain the distribution equilibrium of the training dataset. In this work, we have chosen two widely used open-source datasets, RAF-DB and FER2013. As well as resolving the imbalance problem, we present a new, balanced dataset, applying data augmentation techniques and cleaning poor-quality images from the FER2013 dataset. We then conduct a comprehensive evaluation of thirteen different ViT models with these three datasets. Our investigation concludes that ViT models present a promising approach for FER tasks. Among these ViT models, Mobile ViT and Tokens-to-Token ViT models appear to be the most effective, followed by PiT and Cross Former models.

**Keywords:** facial emotion recognition; vision transformer; data augmentation; balanced data; FER2013; RAF-DB

## 1. Introduction

Facial expression recognition is a key subfield of human–computer interaction, impacting fields such as sentiment analysis, affective computing, and virtual reality. Interest in cutting-edge techniques that improve accuracy and resilience is growing as there is a greater requirement for robots to recognize and respond to human emotions appropriately. Transformer-based models, which were initially developed for language problems, bring in a new era of performance for various image classification tasks in computer vision. This work launches an investigation, examining various vision transformer models in the context of recognizing emotions, and supported by facial datasets.

At the core of our study lies a close look at a range of vision transformer architectures. Our main goal is to comprehend how accurately these structures represent the subtleties of facial expressions. We also investigate how data augmentation techniques enhance model performance, particularly in datasets with balanced classes, to further the depth of our analysis. The FER2013 dataset, known as a benchmark repository containing the complete range of human emotional expressions, serves as the foundation for our empirical inquiry. This dataset has several limitations, such as an imbalance between classes and low-quality images. These shortcomings can negatively affect the performance of models trained using this dataset. Therefore, we developed a new, balanced dataset based on FER2013 that will address these deficiencies.

An overview is given in Section 2, which also explores the field of facial expression recognition, the idea of transformer architecture, and the importance of contemporary data-augmentation approaches. The description of our methodology in Section 3 includes information regarding the vision transformer models. We present the findings of our studies in Section 4, analysing the subtleties of each model's performance.

## 2. Related Works

### 2.1. Facial Emotion Recognition

The recognition of facial emotions has captured substantial attention due to its pivotal role in applications like human–computer interaction and affective computing. In this field, various techniques have been investigated. Early approaches frequently used manually created features and ML techniques such as support vector machines and random forests. Convolutional neural networks have shown their strength in capturing detailed spatial patterns within facial emotions as deep learning approaches have grown in popularity [1].

### 2.2. Transformer Models in Computer Vision

A paradigm change has occurred as a result of the introduction of transformer models—originally developed for natural language processing—to the field of computer vision. The principles of attention and multi-head attention described in the original transformer paper (presented in Figure 1) must be understood before delving deeply into how vision transformers function. In [2], the transformer model is put forth. Three variables are used in the transformer's attention mechanism: K (Key), Q (Query), and V (Value). As inputs, the sets of key-value pairs with query vectors are used. The softmax operator is used to calculate the output vector after a weighted sum of the values is determined; the weights are determined using a scoring function (Equation (1)).

$$\text{Attention}(Q, K, V) = \text{softmax}\left(\frac{Q{\cdot}K^{T}}{\sqrt{d_{k}}}\right){\cdot}V \tag{1}$$

where V, Q, and $K^{T}$ are value, query, and transposed key matrix, respectively. The scaling factor is $1/\sqrt{d_{k}}$, and $d_{k}$ represents the key matrix's dimensions.

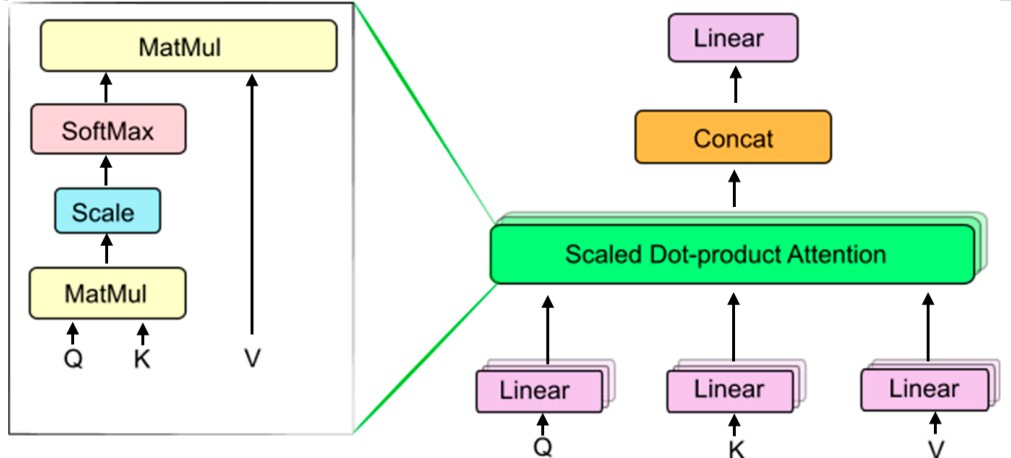

**Figure 1.** The attention layer's structure. Scaled dot-product attention is on the left, multi-head attention mechanism on the right.

The multi-head attention mechanism is defined using the Q, K, and V calculation as a single head. Q and K are used by the (single head) attention mechanism in the preceding diagram.

However, in the multi-head attention mechanism, each head has its own projection matrix, and they use the feature values projected using these matrices to determine the attention weights.

$$
\begin{aligned}
\text{Multihead}\left(Q', K', V'\right) &= \text{Concat}\left(\text{head}_1, ..., \text{head}_h\right)W^O \\
\text{where head}_i &= \text{Attention}\left(QW_i^Q, KW_i^K, VW_i^V\right)
\end{aligned}
\tag{2}
$$

where, $Q', K', V'$ are the projection matrix; $W_{1...h}^Q \in \mathbb{R}^{D \times d_k}$, $W_{1...h}^K \in \mathbb{R}^{D \times d_k}$, $W_{1...h}^V \in \mathbb{R}^{D \times d_v}$ and $W^O \in \mathbb{R}^{h \cdot d_v \times d_{out}}$ are trainable weight matrices; D is the size of the embedding vector of each input element from our sequence; $d_k$ and $d_v$ are the inner dimensions of each self-attention layer, and h is the number of heads.

Multi-head attention is theoretically based on the ability to consistently pay attention to different sequence pieces in diverse ways. The model can better collect location information as a result because, in practice, each head will focus on a separate input segment. We will have a stronger representation as a result of their combination. Each head will also uniquely correlate words to gather various pieces of contextual information.

A turning point was reached with the development of the vision transformer (ViT) [3], which highlighted the potential of transformer designs in image categorization. With this method, each patch of an image is processed by a layer of a transformer. This novel viewpoint cleared the way for the successful representation of global contextual information in images.

The following are some significant application areas for vision transformers, which are widely used in common computer vision tasks: image classification, segmentation, object detection, and cluster analysis.

CrossViT was suggested by the authors in [4] for image classification. Excellent teacher-guiding small networks (ES-GSNet) were suggested by the authors in [5] for the classification of remote sensing image scenes. The authors of [6] provided additional details on the application of ViT for the multilabel classification of satellite imagery and suggested ForestViT. ViT was used by Tanzi et al. in [7] to classify femur fractures.

With regard to detection, Beal et al. [8] were the first to use a supervised pretrained ViT in conjunction with a faster region-based convolutional neural network detector for object detection. The authors of [9] suggested an unsupervised learning-based method for identifying manipulation in satellite photos by the use of ViT. A bridged transformer (BrT) was suggested by the authors in [10] for 3D object detection. The model was used for 3D object detection in point clouds and vision.

Transformers can also be used to segment images. Medical image segmentation was accomplished in [11] using a combination of ViT and U-Net. The transformer was used by the authors in place of the encoder in the traditional U-Net. The authors of [12] presented a brand-new image segmentation technique called "language-aware ViT" (LAVT). In a similar vein, high-resolution ViT for semantic segmentation was proposed in another work [13]. MaskFormer was proposed by Cheng et al. in [14] for image segmentation.

As to image super-resolution, Eformer was suggested by Luthra et al. in [15] for medical image denoising. The authors proposed SUNet for image denoising in [16] by way of combining UNet and the Swin transformer [17]. The authors of [18] suggested DenSformer as a method for image denoising. The three modules that made up the DenSformer were preprocessing, feature extraction, and reconstruction.

Anomaly detection is another potential use. In [19], a unique ViT network for image anomaly detection and localization was created. The authors of this study used the BTAD real-world dataset. In a similar vein, the authors of [20] suggested AnoViT for anomaly detection and localization. TransAnomaly is a video ViT and U-Net-based framework that Yuan et al. proposed in [21] for the detection of anomalies in the videos.

The results show that the main applications of ViTs are as follows: 50% are for image classification, 40% are for object detection, 1% are for segmentation, 1% are for compression, 2% are for super-resolution, 3% are for denoising, and 3% are for anomaly detection [22].

### 2.3. Attaining Data Balance through Augmentation

In the pursuit of building resilient machine-learning models, tackling data imbalance assumes a pivotal role. Researchers have used data augmentation techniques as an effective tactic in the field of recognizing facial emotions. These methods cover a wide range of modifications, from colour adjustments and random erasing to geometrical tweaks like rotation and scaling [23]. By using these augmentation strategies, the dataset's class distribution is made more equitable, which improves model performance and resilience.

The foundation of our work is the integration of several interrelated research fields—computer vision transformer models, facial emotion recognition, and data augmentation. Our research focuses on using vision transformers' potential to improve the precision of emotion recognition. We aim to push the limits of accurate emotion recognition by utilising the synergies between these domains.

## 3. Analysis of Vision Transformer Models

The use of vision transformers is widespread in common tasks such as object detection, image segmentation, image classification, and action recognition. In the present research, we used vision transformer (ViT) models to tackle the issue of identifying facial emotions. This part introduces the topic by examining the fundamental mechanics and architectural features that these models have, as well as illuminating the conceptual framework of ViT models.

### 3.1. Vision Transformer Models

3.1.1. Base Vision Transformer (ViT)

An innovative model that expands the transformer architecture to handle image classification problems was introduced by Dosovitskiy et al. [3] in 2020. ViT takes a distinctive approach to image processing, much like its equivalent in natural-language processing. The typical transformer model used a one-dimensional sequence of word embeddings as input due to the model design, which was based on NLP. On the other hand, when the transformer model is used for the problem of image classification in CV, the input data is given to it as two-dimensional images. Figure 2 presents an illustration of the main ViTs.

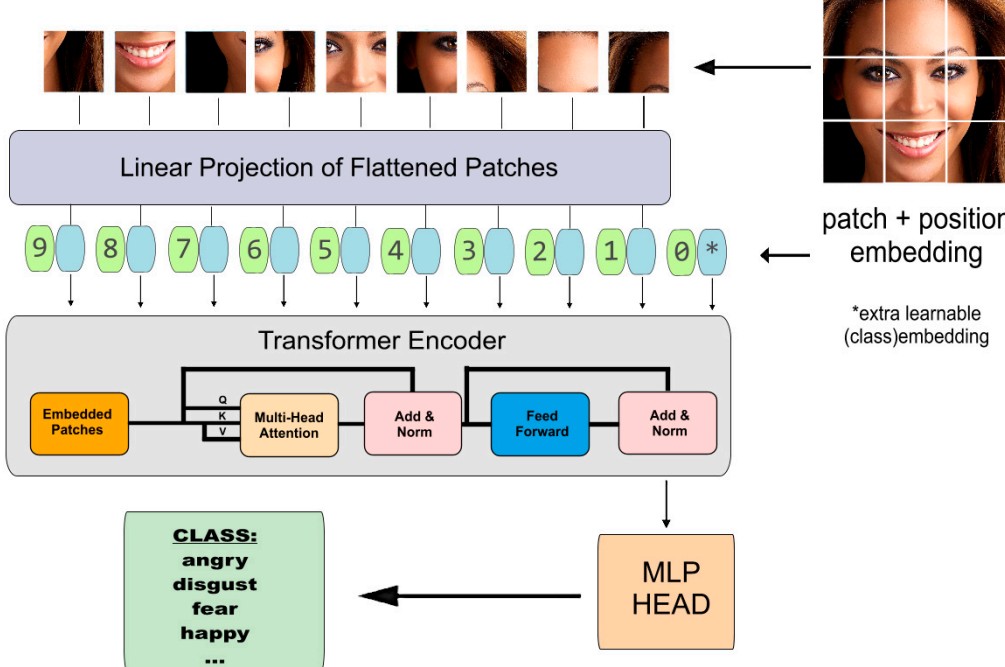

**Figure 2.** Architecture of the base vision-transformer model.

Divided into smaller two-dimensional patches, the input image, with height, width, and number of channels, is used to structure the input image data in a manner consistent with the input's structure in the NLP domain. As a result, there are $N = \frac{HW}{P^2}$ patches, with each patch having a resolution of (P, P) pixels.

The following operations are carried out prior to feeding the data into the transformer encoder:

- Every picture patch is flattened into an $x_p^n$ vector of length $P^2 \times C$, where $n = 1, \ldots N$.
- Mapping the flattened patches to D dimensions using a trainable linear projection, E, produces a series of embedded picture patches.
- The classification output, y, is represented by the learnable class embedding, $x_{class}$, which prefixes the list of embedded image patches.
- Finally, positioning information, which is also learned during training, is added to the input by augmenting the patch embeddings with one-dimensional positional embeddings $E_{pos}$.

The embedding vectors that arise from the aforementioned procedures are as follows:

$$z_0 = \left[ x_{class}; x_p^1 E; \ldots ; x_p^N E \right] + E_{pos} \qquad (3)$$

where, $E_{pos}$ is the positional embedding matrix, $x_p^n$ is a flattened vector of image patches, and $x_{class}$ is the class embedding vector.

At the input of the transformer encoder, a stack of L identical layers $z_0$ is fed for classification. They then proceed to feed a classification head with the value of $x_{class}$ at the $L^{th}$ layer of the encoder output.

It is crucial to understand the transformer encoder and its essential components. These are the parts of the transformer encoder:

- Multi-head self-attention (MHSA) layer: By using multiple "heads", the ViT model can simultaneously focus on different segments of an image. Each head calculates attention independently, allowing for a variety of image representations to be generated. These representations are then combined to create a final image representation. This approach allows the model to capture more nuanced interactions between input components. However, this also makes the model more complex and computationally expensive due to the need to aggregate the outputs from all the heads. Within the images in Figure 3, we offer insights into the visualization of the attention mechanism in a vision transformer. We explore different numbers of heads in the multi-head self-attention layer. The upper-left corner displays the original image and its attention visualization using the mean value of the heads. Meanwhile, the lower part of the image showcases visualizations with varying numbers of heads in the multi-head self-attention layer. Based on Figure 3, it is evident that as we increase the number of heads in the MHSA layer, the model's ability to identify interrelated objects in our dataset improves. However, it is worth noting that using a large number of heads can also negatively impact the accuracy of the model. Therefore, in most cases, it is crucial to determine the optimal number of heads for each model. In Figure 3, the ideal choice would be to set the number of heads to 4.
- Layer normalization (LN): LN is used to normalize the training data before each block, preventing the introduction of any new dependencies. This improves overall performance and training effectiveness.
- Feed-forward network (FFN): The MHSA layers produce outputs that are processed by the FFN. It has a nonlinear activation function and two linear transformation layers.
- Multi-layer perceptron: This layer uses the GELU activation function in a two-layer structure.

To sum up, ViT utilizes the encoder part of the transformer framework. The encoder takes in a series of embedded image patches, along with positional information and a learnable class embedding added at the start of the sequence. The learnable class embedding

value is sent to a classification head linked to the encoder's output, which utilizes it to generate a classification outcome based on its condition.

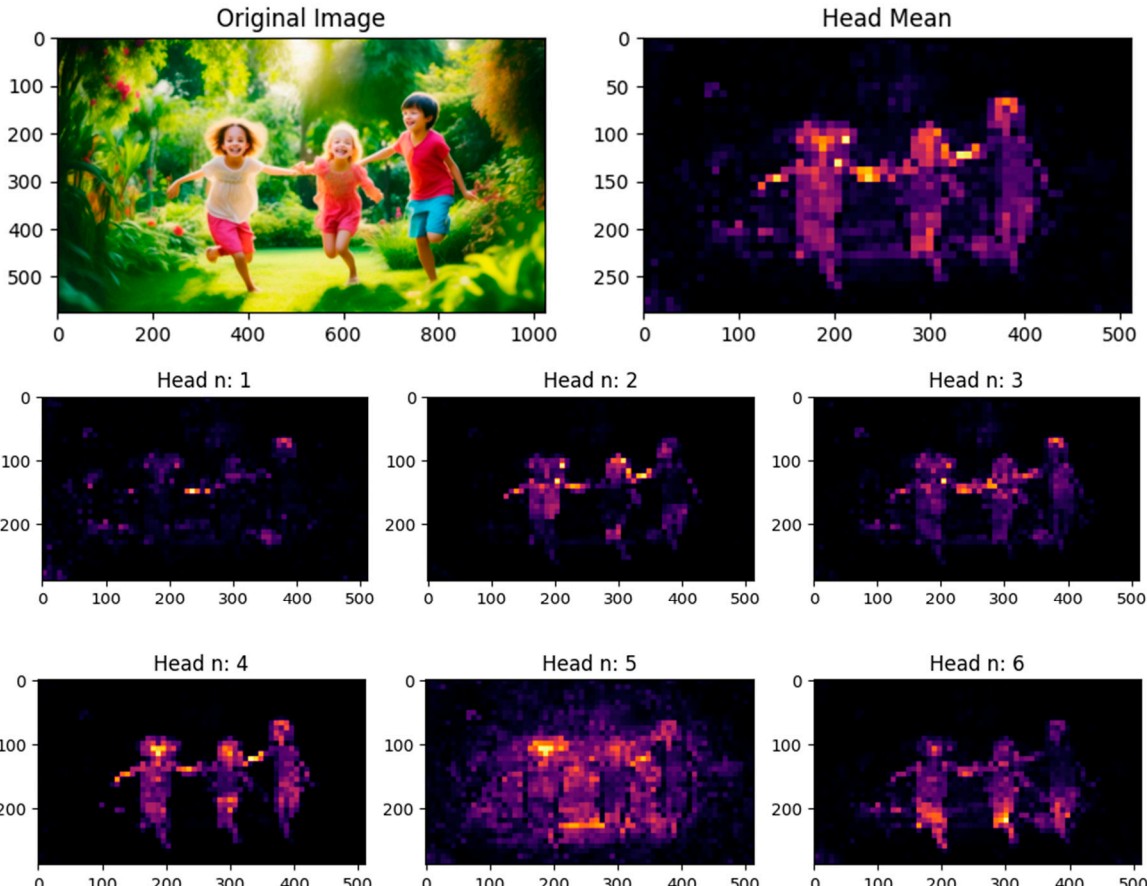

**Figure 3.** Attention maps in a vision transformer (ViT) with varying head numbers.

### 3.1.2. Attention-Based Approaches

To capture various features of the image, a number of models use several ViTs operating in tandem. A self-attention module and a class attention module, for instance, are used in the CaiT [24] architecture. While the class attention module learns the connections between the patches and the class labels, the self-attention module focuses on the connections between the patches themselves. The model can learn both the local and global aspects of the image using this method. Models such as ViT for Small-Size Datasets [25], Deep ViT [26] and Cross Former [27] also employ attention-based strategies.

### 3.1.3. Patch-Based Approaches

Other ViT architectures focus on improving the segmentation of images into smaller sections. One example is the Tokens-to-Token vision transformer (T2T-ViT) [28] architecture, which creates patches in an iterative manner using a token-to-token module, as shown in Figure 4. This allows the model to capture patches of different sizes, which can be helpful in capturing varying levels of image detail.

### 3.1.4. Multi-Transformer-Based Approaches

Some vision transformer architectures improve the efficiency and effectiveness of the ViTs' self-attention module. A cross-attention module, for instance, is used in the CrossViT [4] architecture (given in Figure 5) to allow information to be transferred across the two branches of the model, one of which processes small patches, and the other, large patches. This approach reduces the computational cost of attention while still allowing

the model to learn long-range dependencies. An approach similar to the CrossViT is also utilized in the Parallel ViT model [29].

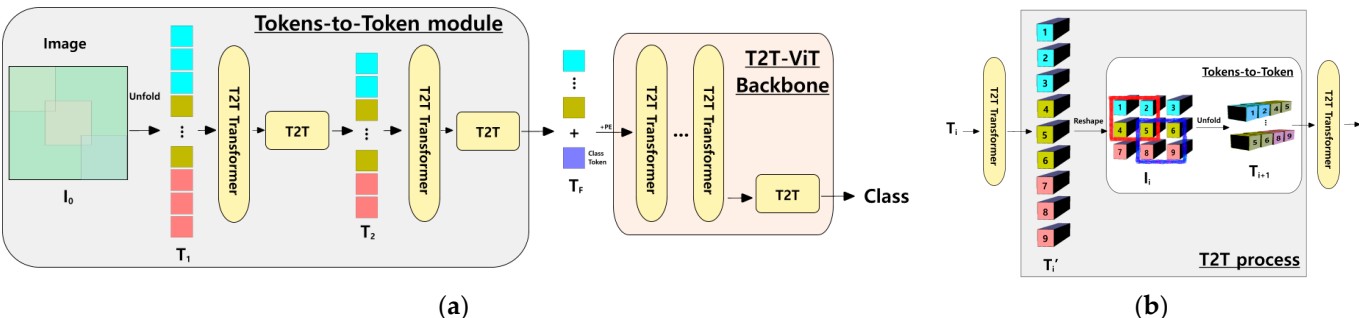

**(a)**                                                          **(b)**

**Figure 4.** Tokens-to-Token vision transformer (T2T-ViT): (**a**) architecture of the Tokens-to-Token vision transformer (T2T-ViT) model; (**b**) schematic representation of the Tokens-to-Token (T2T) process.

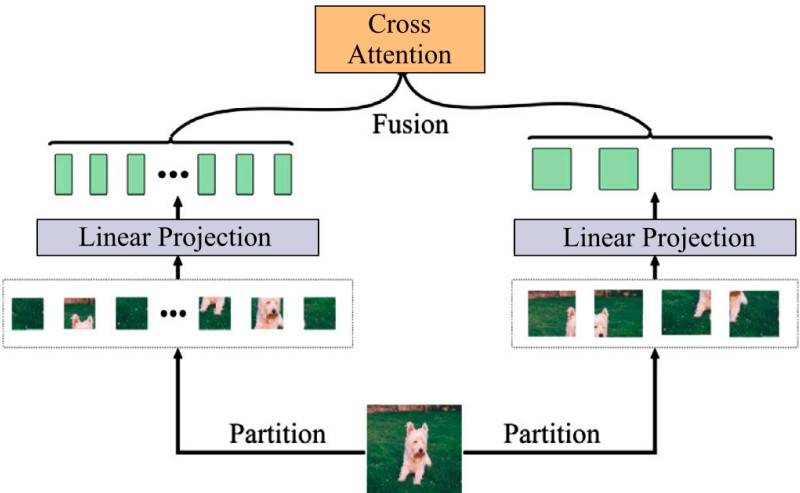

**Figure 5.** The general design of the basic cross-attention mechanism.

### 3.2. Hybrid Vision Transformer Architectures

Vision transformer differs from conventional convolutional neural networks (CNNs) in several ways, including its capacity to learn long-range dependencies in the image and its training on very large datasets. Although ViTs have certain drawbacks, such as high computing costs and a propensity to disregard geographical data, researchers have proposed several hybrid vision transformer (HVT) architectures that combine CNNs and transformers to address these issues.

To grasp long-distance relationships, the LeViT [30] design utilizes a CNN to transfer the local features extracted from the image to a ViT encoder. This approach allows the model to promptly capture spatial data from the picture, improving its ability to execute tasks such as object detection and segmentation. As well, several HVT models, including EarlyConViT [31], Mobile ViT [32], Region ViT [33], and PiT [34] are integrated with CNN.

We investigated the use of facial emotion recognition with a variety of vision transformer (ViT) models, including those mentioned above.

## 4. Experimental Results

As described in this section, we conducted a comprehensive experiment to test vision transformer models on facial expression databases. First, we provide a brief overview of the databases used in our research.

*4.1. Datasets*

We used two public facial expression recognition (FER) datasets (RAF-DB and FER2013) and our new Balanced FER2013 dataset to showcase the performance of vision transformer models across different datasets.

4.1.1. FER2013 Dataset

The FER2013 dataset (Figure 6), known for its wide range of facial expressions, serves as the foundation for our empirical investigation. FER2013 consists of a total of approximately 35,887 images.

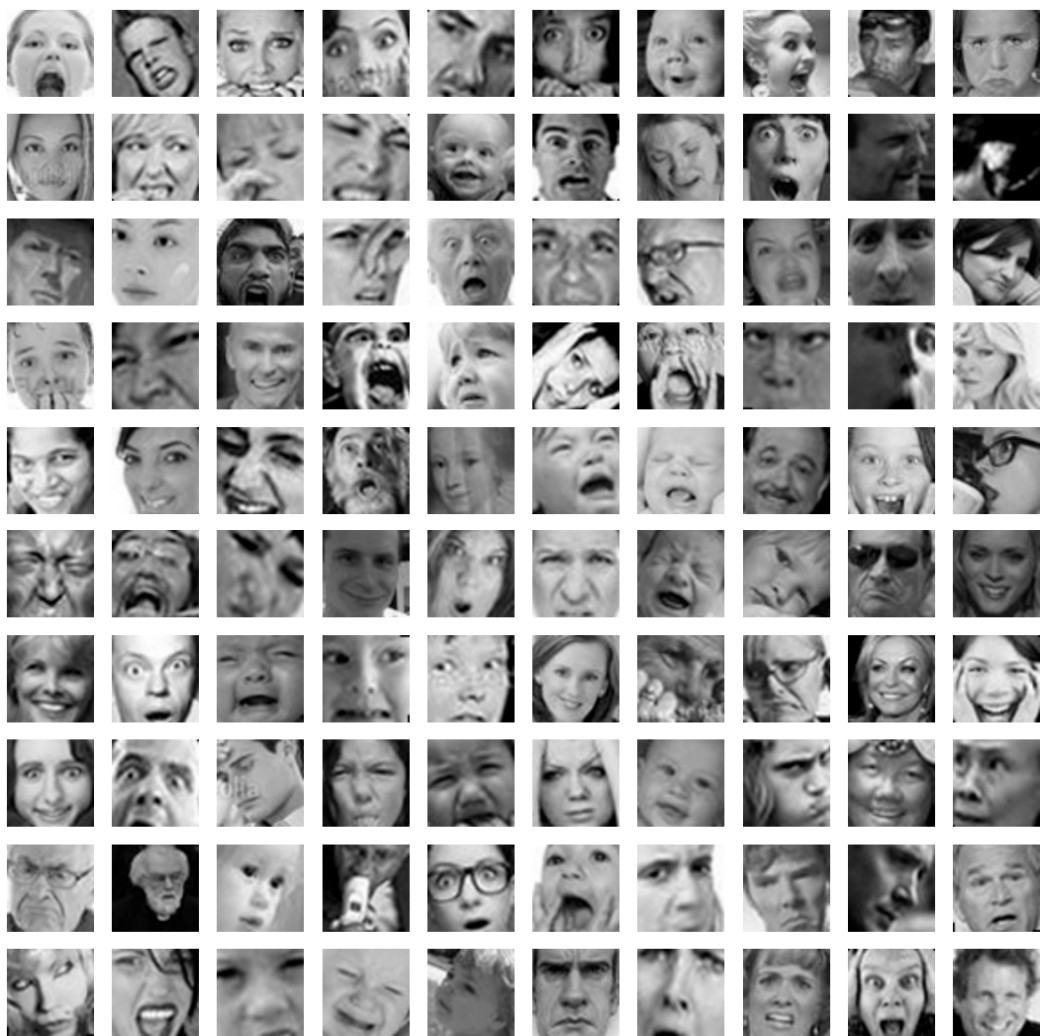

**Figure 6.** FER2013 dataset samples.

This total includes images from the training, public-test, and private-test subsets combined. The dataset includes seven different emotion categories—anger, disgust, fear, happiness, sadness, surprise, and neutral—and offers a solid framework for assessing the recognition abilities of various ViT models.

4.1.2. RAF-DB Dataset

The Real-World Affective Faces Database (RAF-DB) is an FER dataset (Figure 7) that includes 29,672 facial images that have been annotated by 40 different taggers describing simple and complex expressions. Facial photos can indicate seven basic emotions and eleven complex emotions. The subject's age, gender, ethnicity, and head posture, as well

as the lighting, occlusions, and post-processing techniques, are all highly variable in the photographs in the RAF-DB.

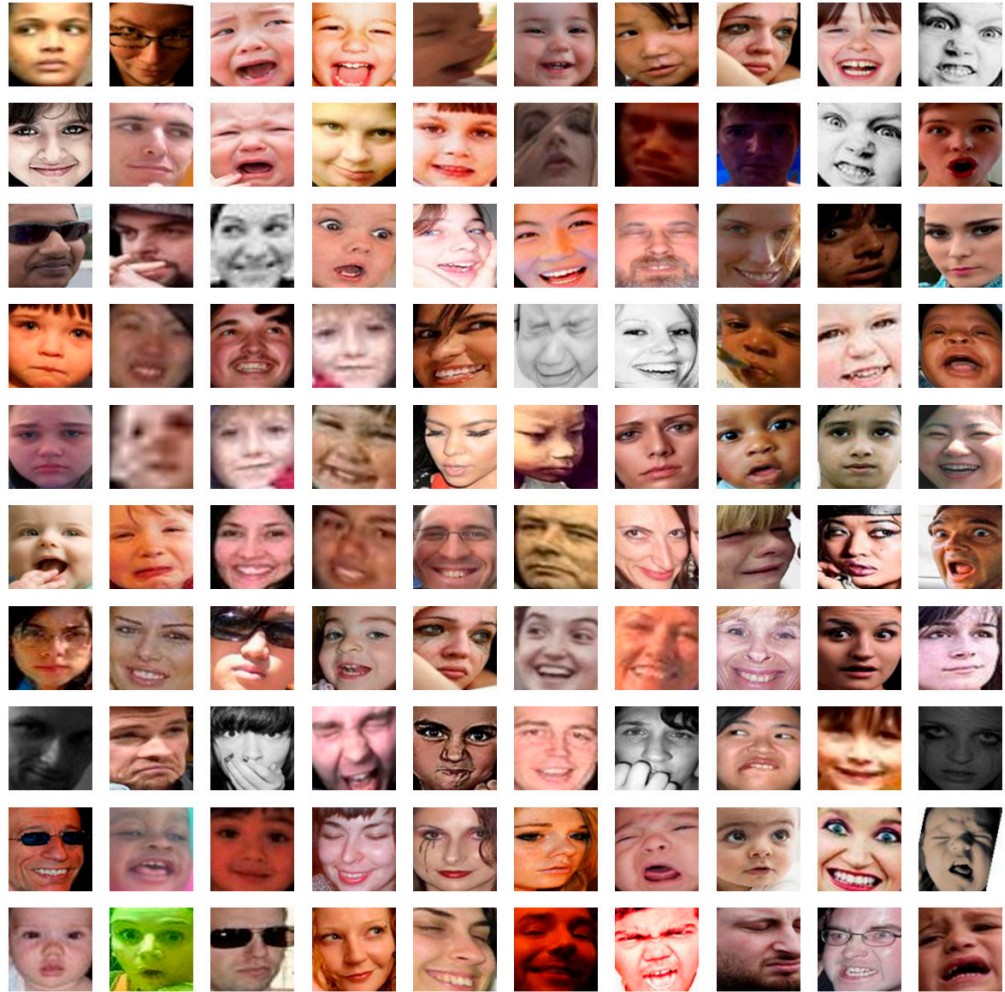

**Figure 7.** RAF-DB dataset samples.

### 4.1.3. Our Balanced FER2013 Dataset

As in other domains, one of the most crucial elements in enhancing the precision of facial emotion recognition models is a balanced dataset. In this study, we used the cleaning and augmentation techniques to build a new, balanced FER2013 dataset, which was based on the existing FER2013 dataset. The FER2013 dataset is a well-known benchmark for facial emotion recognition (FER) tasks. But there are some issues with the FER2013 dataset. Among the main challenges are:

- Contrast variation: It is possible that the dataset contains photos that are either too bright or too dark. CNN models, which learn visual features automatically, tend to perform better with high-contrast images. Low-contrast images, on the other hand, may affect CNN performance due to the lower amount of information they transmit. This issue can be resolved by improving the quality of the faces in the pictures.
- Imbalance: When one class has many more photos than another, there is a class imbalance. This may skew the model in favour of the dominant class. The model will favour the cheerful class, for instance, if there are 100 photographs of joyful people and 20 images of afraid people. To address this issue, data augmentation techniques like horizontal flipping, cropping, and padding can be applied to increase the amount of data available for the minority classes [35].

- Intra-class variation: A range of facial expressions, including animated faces and drawings, are included in the dataset. The features of real and animated faces differ, which can make it challenging for the model to extract landmark elements. In order to enhance model performance, only photographs of actual human faces should be included in the dataset.
- Occlusion: When part of the image is hidden, this is known as occlusion. This may occur when someone is wearing sunglasses or a mask, or when a hand covers a portion of the face, like the right eye or nose. Occluded photos should be eliminated from the dataset, since the eyes and nostrils are crucial characteristics for detecting and extracting emotions.

The FER2013 dataset can be enhanced to become a more trustworthy standard for FER research by resolving these issues. In this work, we present a new dataset, FER2013_balanced, which is obtained by data augmentation methods, using the FER2013 dataset. Initially, we excluded poor-quality photos from the FER2013 dataset, including those with low contrast or occlusion. A depiction of poor-quality images from the FER2013 dataset is shown in Figure 8, arranged according to the relevant categories.

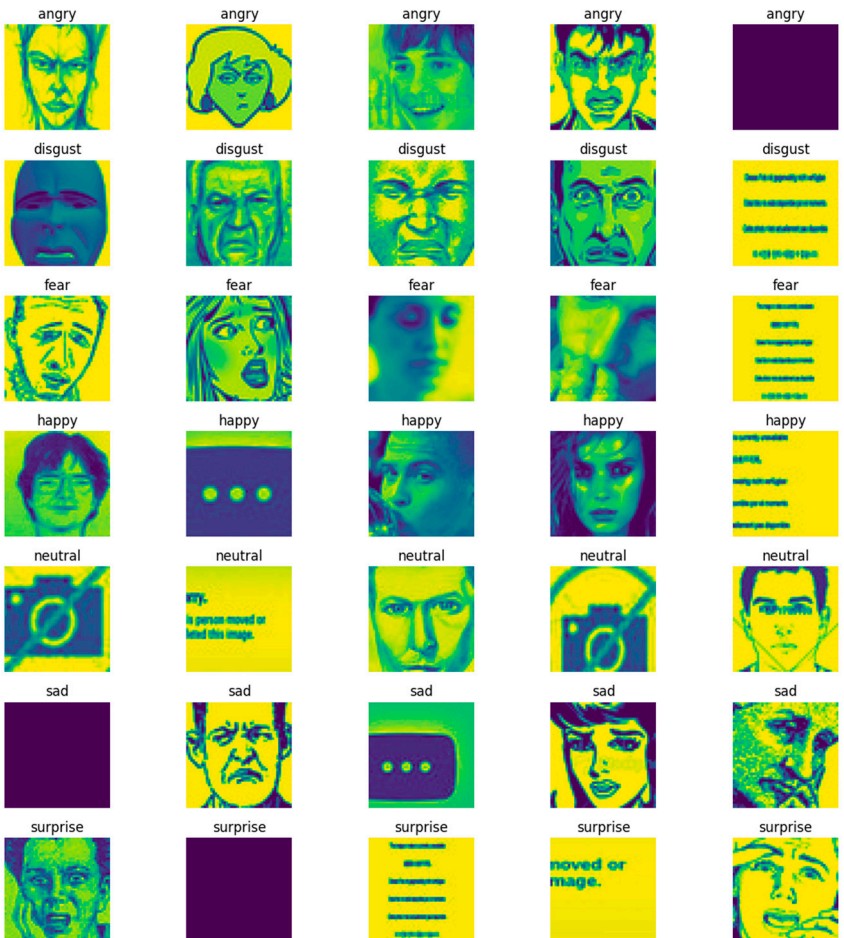

**Figure 8.** Examples of poor-quality images sourced from the FER2013 database.

Next, in order to enlarge the minority classes, we employed data augmentation techniques on the remaining images. Finally, we eliminated a few photos from the happy, neutral, sad, and other categories to balance our dataset. The FER2013_balanced dataset contains an equal number of photos for every emotion category. This ensures a balanced distribution of classes, reducing the possibility of bias towards the majority classes. Thus, it serves as a reliable baseline for FER research. The number of images in each category from the FER2013 and FER2013_balanced datasets is shown in Figure 9.

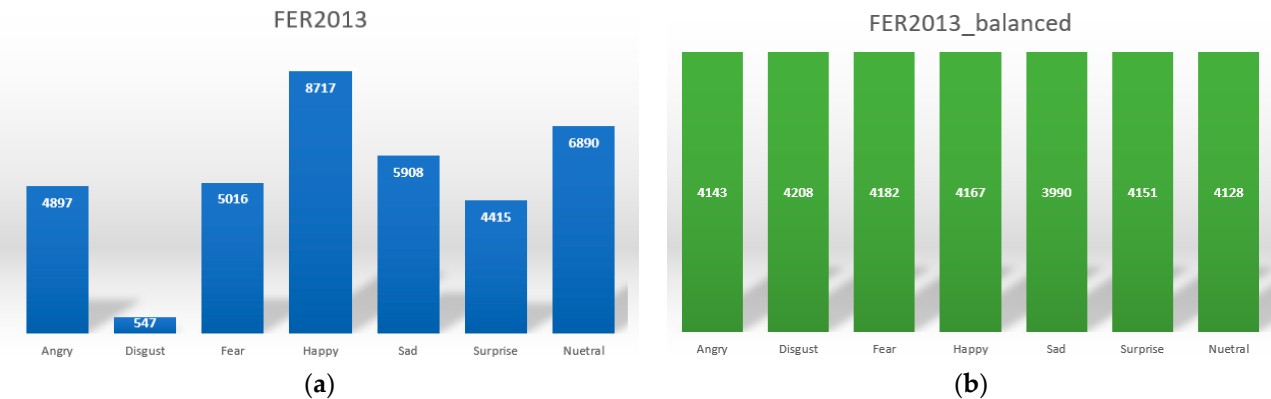

**Figure 9.** Class-wise image distribution in the FER2013 dataset: (**a**) the image distribution per class in the original FER2013 dataset; (**b**) the image distribution for the FER2013 balanced dataset.

### 4.2. Results

In this section, we will discuss the results of our evaluation and comparison of 13 different vision transformer (ViT) models on the facial expression recognition RAF-DB and FER2013 datasets, as well as our Balanced FER2013 dataset. Our process involved training the model on a portion of the dataset, testing it on a validation set, and then reporting its accuracy on the test set. Before we describe the specific aspects of the model's performance on various datasets, we will briefly outline our training process. We ensured consistency in the architecture and hyperparameters of each model and trained a single model for each dataset. For all experiments, we utilized a workstation featuring an Intel i7 CPU with 32 GB of RAM and an Nvidia GTX 3090 GPU with 24 GB of memory. In this study, each vision transformer model was trained for 25 epochs from scratch to develop the models. To optimize the model, we applied an Adam optimizer with a learning rate of $3 \times 10^{-5}$. In order to ensure that the model was trained on a diverse range of images and could handle minor variations, we applied data augmentation to the images in the training sets. This involved flipping, zooming, rotating, and distorting the images to enhance the quality of the data.

We measured each model's performance in terms of recognition accuracy. The following Table 1 provides an overview of the accuracy results obtained:

**Table 1.** Classification accuracy (%) of different vision-transformer models on facial expression recognition datasets.

| Model | RAF-DB | FER2013 | Balanced FER2013 |
|---|---|---|---|
| Base ViT [3] | 68.34 | 49.64 | 60.25 |
| CrossViT [4] | 69.74 | 50.27 | 62.43 |
| CaiT [24] | 70.45 | 45.68 | 60.15 |
| ViT for Small-Size Datasets [25] | 72.37 | 55.35 | 67.88 |
| Deep ViT [26] | 63.25 | 43.45 | 50.37 |
| Cross Former [27] | 72.47 | 59.95 | 75.12 |
| Tokens-to-Token ViT [28] | 76.40 | 61.28 | 74.20 |
| Parallel ViT [29] | 67.16 | 50.94 | 64.40 |
| LeViT [30] | 65.71 | 47.22 | 60.85 |
| Early ConViT [31] | 68.24 | 51.02 | 66.70 |
| Mobile ViT [32] | 74.28 | 62.73 | 77.33 |
| Region ViT [33] | 69.62 | 56.03 | 73.79 |
| PiT [34] | 72.84 | 58.67 | 76.09 |

The Tokens-to-Token ViT model outperforms all other models on the RAF-DB dataset, achieving accuracy values of 76.40%. With high accuracy rates across all three datasets, Mobile ViT, PiT and Cross Former models also produce great results. The Deep ViT and

Parallel ViT models underperform on all three datasets. They may not be suitable for FER, as they were developed for other tasks and may not work efficiently with small databases.

Further observations indicate that the models perform insufficiently on the FER2013 dataset compared to the RAF-DB dataset. The reason for this could be the challenging nature of the images in the FER2013 dataset, which exhibit higher levels of noise and occlusion. However, on the Balanced FER2013 dataset, the models perform better than on the FER2013 dataset. This is because the Balanced FER2013 dataset has an equal number of images for each expression class, while the FER2013 dataset is imbalanced.

In Figure 10, one can compare each emotion category's accuracy with the Tokens-to-Token ViT model's application to the FER2013 and FER2013_Balanced datasets. As can be seen from the following categories, the model achieves high accuracy when using the FER2013_balanced dataset. The analysis covers a broad range of emotions including anger which increased from 47.3% to 73.4%, disgust which improved from 43.5% to 91.2%, fear which rose from 33.5% to 79.1%, and neutral which advanced from 60% to 79%. However, it is also important to acknowledge a few negative outcomes in the categories of surprise, sadness, and happiness. When it comes to overall accuracy of the Tokens-to-Token ViT, the FER2013 dataset has an accuracy of 61.28%, whereas the FER2013_balanced dataset has an accuracy of 74.20%.

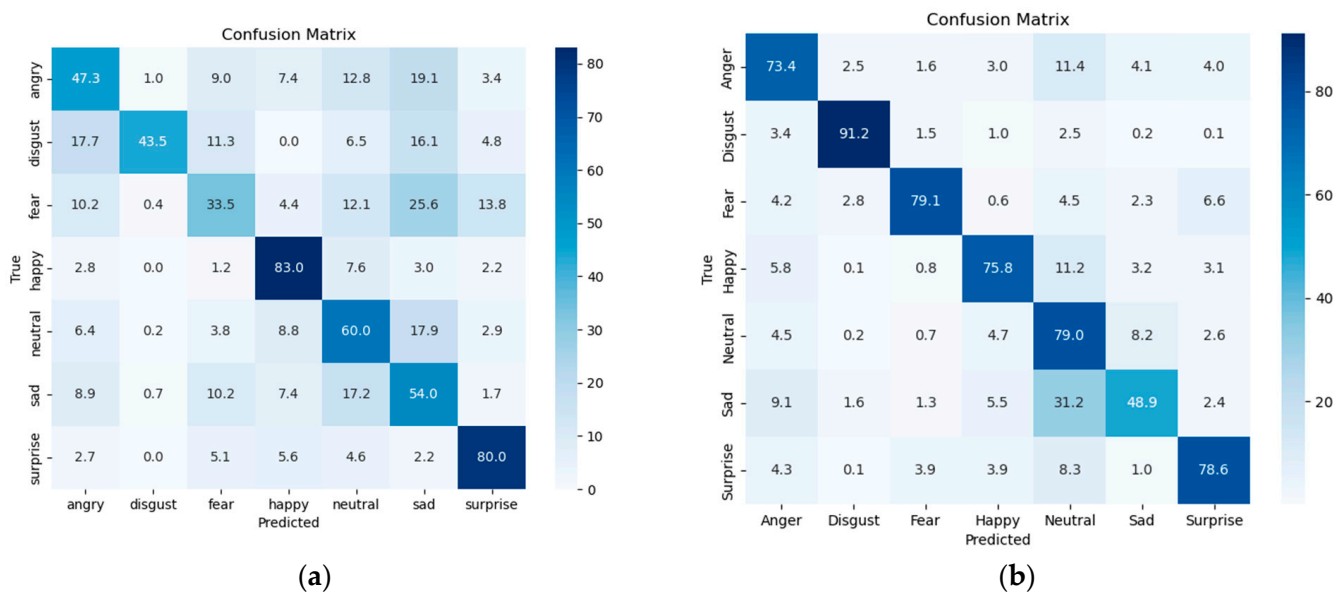

**Figure 10.** The confusion metrics of Tokens-to-Token ViT [28] model: (**a**) on the FER2013 dataset; (**b**) on the FER2013_balanced dataset.

In terms of facial expression recognition (FER), the Mobile ViT and Tokens-to-Token ViT models are the most effective ViT models, followed by the PiT and Cross Former models. Conversely, the Deep ViT and Parallel ViT models showed subpar performance. The Base ViT model, however, proved to be a decent baseline model for this task.

## 5. Conclusions

In this study, we used three datasets, the RAF-DB, the FER2013, and our Balanced FER2013 Facial Expression Recognition dataset, to carry out a thorough investigation of vision transformer (ViT) models for facial emotion recognition. Our research shows that the Tokens-to-Token ViT and Mobile ViT models are more effective than the PiT and Cross Former models for FER.

Our findings also suggest that ViT models can be successful in completing FER tasks, even with challenging datasets like FER2013. However, to improve the performance of ViT models on FER datasets with noise and occlusion, further research is needed.

Our study provides valuable insights into the use of ViT models for FER. The results can assist other researchers in selecting the most suitable ViT model for their FER work and in identifying areas that require further exploration.

**Author Contributions:** Conceptualization, S.B. (Sukhrob Bobojanov) and B.M.K.; methodology, S.B. (Sukhrob Bobojanov), B.M.K. and M.A.; software, S.B. (Sukhrob Bobojanov) and S.B. (Shohruh Begmatov); validation, S.B. (Sukhrob Bobojanov), M.A. and S.B. (Shohruh Begmatov); formal analysis, B.M.K., S.B. (Sukhrob Bobojanov), M.A. and S.B. (Shohruh Begmatov); investigation, S.B. (Sukhrob Bobojanov) and M.A.; data curation, S.B. (Sukhrob Bobojanov), M.A. and S.B. (Shohruh Begmatov); writing—original draft preparation, S.B. (Sukhrob Bobojanov), M.A. and S.B. (Shohruh Begmatov); writing—review and editing, S.B. (Sukhrob Bobojanov), B.M.K. and M.A.; visualization, S.B. (Sukhrob Bobojanov) and M.A.; supervision, B.M.K.; project administration, S.B. (Sukhrob Bobojanov) and B.M.K.; funding acquisition, S.B. (Sukhrob Bobojanov). All authors have read and agreed to the published version of the manuscript.

**Funding:** This research received no external funding.

**Institutional Review Board Statement:** Not applicable.

**Informed Consent Statement:** Not applicable.

**Data Availability Statement:** The data presented in this study are available on request from the corresponding author. The data are not publicly available due to ongoing project.

**Conflicts of Interest:** The authors declare no conflict of interest.

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
