# Peer review of "Comparative Analysis of Vision Transformer Models for Facial Emotion Recognition Using Augmented Balanced Datasets"

_applsci, doi:10.3390/app132212271_

Round 1

Reviewer 1 Report

Comments and Suggestions for Authors

In this review paper, the authors evaluate 13 Vision Transformer models on three facial emotion recognition datasets. They find that Tokens-to-Token ViT model has the best performance, followed by PiT and Cross Former models. My concerns are listed in the following.

  1. Please indicate the meaning of d_k in Eq. 1.
  2. How do you balance the original FER2013 to the new balanced FER2013 dataset? Could you explain your balance method and procedure in details?
  3. Is RAF-DB a balanced dataset? Did you try to balance it to improve the performance?
  4. Is there any way to dive into the different performance of different models on the datasets? Could you give more analysis of the different performance on different datasets?
  5. From the results, it seems like that Mobile ViT has the best performance. Why do you state that T2T ViT has the best performance?

Author Response

Dear Reviewer

Reviewer 2 Report

Comments and Suggestions for Authors

This paper researches the realms of human-computer interaction and sentiment analysis. The authors present their research findings, which are based on an assessment of ViT (Vision Transformers) models using three datasets. Their research primarily centers on leveraging Vision Transformers to enhance emotion recognition precision through the synergistic interplay of computer vision transformer models, facial emotion recognition, and data augmentation.

 The abstract does have a set of results in percentages and a detailed breakdown of these percentages, yet it lacks contextualization regarding the theoretical underpinnings of the research challenges. This includes not clarifying why these specific datasets were chosen and perhaps providing a brief explanation of the significance of enhanced balanced datasets. Furthermore, there's room for improvement in terms of the methodological approach, which needs more clarity and substantiating evidence on the research process. The current presentation of consecutive figures without intervening explanations or paragraphs is not aligned with best practices for scientific research papers.

Line 34-35: please review because it is too vague/generic, and needs to be more objective with fewer activations (e.g., crucial – is mentioned twice)

Line 41: should consider resuming the understanding about the relevance of using enhanced balanced datasets; what differentiates these datasets from the other datasets?

Line 50: review the need to keep the first sentence (no added value)

Line 70: consider removing and replacing it with the author's perception/understanding that is supported or explored deeply in the cited article. The IEEE citation format is a sequential numbering, jumping from citation [2] to citation [28] is inconsistent.

Line 80: Figure 1 is not explained in the paper. Not acceptable that none of the 14 Figures are mentioned or described in the paper. What is the relevance or role of a figure with a legend that is not described or analyzed within the paper?

Line 97-99: too generic, it does not provide any useful information to the reader, missing explaining the architectural modifications introduced by the listed Image Transformers techniques and why those techniques are relevant. The Related Works chapter should be revised 

Comments on the Quality of English Language

Moderate editing of English language required

Author Response

Dear Reviewer

Reviewer 3 Report

Comments and Suggestions for Authors

This is a pure experimental study on facial emotion recognition using various existing Vision Transformers.  The topic suits the special issue. There are a number of comments to be addressed in the revision.

 1.   Besides image processing, transformer is also used in video processing, e.g., [R1]. The authors should mention more applications of transformers in visual data research.

2.   Why Figure 5 appears earlier than Figure 4 ? Also, in the text, there is no detailed description of Figures 4 and 5.

3.   The dataset being used seems to have already found the face. It is very different from the scenario of Figure 3, where there may be multiple faces, and the deep learning algorithm needs to first local where are the faces. Could the authors explain why the vision transformer is useful in emotion recognition, similar to that in Figure 3 ?

4.   Besides comparing existing vision transformers for facial emotion recognition, is there any original invention from this paper ?

[R1] `MGFN: Magnitude-Contrastive Glance-and-Focus Network for Weakly-Supervised Video Anomaly Detection," AAAI 2023.

Author Response

Dear Reviewer

Round 2

Reviewer 2 Report

Comments and Suggestions for Authors

This paper researches the realms of human-computer interaction and sentiment analysis. The authors present their research findings, which are based on an assessment of ViT (Vision Transformers) models using mainly the comparing of two datasets (the balanced FER2013 dataset and the original FER2013 dataset). Their research primarily centers on leveraging ViT to enhance emotion recognition precision through the synergistic interplay of computer vision transformer models, facial emotion recognition, and data augmentation.

The abstract does not clarify why these specific datasets were chosen; a brief explanation of the significance of an enhanced balanced dataset would be recommended. Furthermore, there's room for improvement in terms of the methodological approach, which needs more clarity and substantiating evidence on the research process.

In this review, the authors have increased the number of adjectives which was precisely one of the critical comments presented in the 1st review. The use of unnecessary adjectives does not contribute to improve the relevance or learnability of the manuscript content. Examples of unnecessary adjectives: 

·        “plays a pivotal role in the context of human-computer interaction”, this is a typical ChatGPT expression;

·        “Facial expression recognition is vital for human-computer interaction”

The current presentation of consecutive figures without intervening explanations or paragraphs is not aligned with best practices for scientific research papers (see fig.10-12, fig. 13-15). No improvement was detected concerning the comments provided in the 1st review regarding these issues.

The authors should reflect on the usefulness of fig. 10-12, eventually replace them with another type of chat that aggregates information with a value proposition to the user's readability. A similar approach should be followed to the information presented in Fig. 13-15.

Line 48: should consider resuming the understanding about the relevance of using enhanced balanced datasets; what differentiates an enhanced balanced dataset from the other dataset? In addition, resume why a balanced class contributes to improving the author's analysis.

Line 84:  missing the meaning of the acronym MHA - Multi-Head 82 Attention; same for NLP, the authors should review the manuscript to avoid having these kinds of issues.

Line 206: Figure 3 should be explained in a paragraph before presenting the figure and not after. Please provide a short (but technical) context to what is presented in the grid of images. The text in lines 197 – 200 is not enough; it requires deeper analysis (more rigor and scientific evidence concerning the applied model or approach).

Line 406: the author's contribution is presented as a list of key concepts/acronyms, no effort in explaining the main research achievements/findings is provided. The conclusions are not supported by the results.

Comments on the Quality of English Language

no comments 

Author Response

Dear Reviewer,

I hope you are doing well

We modified the paper and responded to the comments, according to your last review.

We sincerely hope that you will accept our paper.

Thank you for your time and valuable comments to improve the quality of our paper. 

Reviewer 3 Report

Comments and Suggestions for Authors

This version addressed my concerns satisfactorily.

Author Response

Dear Reviewer,

I hope you are doing well

We do appreciate your trust. In the near future, we plan to write another new paper using the experiences gained from the very professional experts in our research field during the submission and revision of the current paper.

Thank you for your time and valuable comments to improve the quality of our paper.